# Therapy Discontinuation after Myocardial Infarction

**DOI:** 10.3390/jcm9124109

**Published:** 2020-12-19

**Authors:** Łukasz Pietrzykowski, Michał Kasprzak, Piotr Michalski, Agata Kosobucka, Tomasz Fabiszak, Aldona Kubica

**Affiliations:** 1Department of Health Promotion, Collegium Medicum, Nicolaus Copernicus University, M. Curie Skłodowskiej 9 St., 85-094 Bydgoszcz, Poland; michalski.piotr@onet.eu (P.M.); akosobucka@wp.pl (A.K.); aldona.kubica@gmail.com (A.K.); 2Department of Cardiology, Collegium Medicum, Nicolaus Copernicus University, M. Curie Skłodowskiej 9 St., 85-094 Bydgoszcz, Poland; medkas@o2.pl (M.K.); tfabiszak@wp.pl (T.F.)

**Keywords:** adherence, myocardial infarction, angiotensin converting enzyme inhibitors (ACEI), P2Y_12_ receptor inhibitors, statins

## Abstract

The discontinuation of recommended therapy after myocardial infarction predisposes patients to serious thrombotic complications. The aim of this study was a comprehensive analysis of permanent as well as short- and long-term discontinuation of pharmacotherapy, taking into consideration the basic groups of medications and nonadherence determinants in a one-year follow-up in post-myocardial infarction (MI) patients. Material and methods: The study was a single center cohort clinical trial with a one-year follow-up including 225 patients (73.3% men, 26.7% women) aged 62.9 ± 11.9 years. In eight cases (3.6%), the follow-up duration was less than one year due to premature death. The following factors were analyzed: lack of post-discharge therapy initiation; short-term therapy discontinuation (<30 days); long-term therapy discontinuation (≥30 days); and permanent cessation of therapy. The analysis of therapy discontinuation was performed based on prescription filling data. Results: Occupational activity (Odds Ratio (OR) 5.15; 95% Confidence interval (CI) 1.42–18.65; *p* = 0.013) and prior MI (OR 5.02; 95% CI 1.45–16.89; *p* = 0.009) were found to be independent predictors of a lack of post-discharge therapy initiation with P2Y_12_ receptor inhibitors. We found no independent predictors of lack of post-discharge therapy initiation with other medications, whether analyzed separately or together. Age above 65 years (Hazard Ratio (HR)—1.59; 95% CI 1.15–2.19; *p* = 0.0049) and prior revascularization (HR—1.44; 95% CI 1.04–2.19; *p* = 0.0273) were identified as independent predictors of therapy discontinuation. Multilogistic regression analysis showed no independent predictors of the cessation of any of the medications as well as the permanent or temporary simultaneous discontinuation of all medications. Conclusions: The vast majority of post-MI patients discontinue, either temporarily or permanently, one of the essential medications within one year following myocardial infarction. The most likely medication class to be discontinued are statins. Older age and prior cardiac revascularization are independent determinants of therapy discontinuation.

## 1. Introduction

European Society of Cardiology guidelines recommend the implementation of dual antiplatelet therapy (DAPT) for 12 months, angiotensin converting enzyme inhibitors (ACEI) or angiotensin receptor blockers (ARB), beta-blockers and statins [1] in patients after myocardial infarction (MI).

Non-adherence to post-MI therapy is a complex and multidimensional phenomenon precluding the achievement of therapeutic targets [2,3,4,5,6,7,8]. The discontinuation of the recommended post-MI therapy predisposes patients to serious thrombotic events, particularly myocardial infarction, in-stent thrombosis, stroke and death [4,9,10,11,12,13,14].

Studies published so far on adherence in the post-MI setting have described only single aspects of this phenomenon, providing analyses of either its prevalence and consequences or its determinants [9,10,11].

The aim of our study was the comprehensive analysis of the permanent cessation of medication as well as the short- and long-term discontinuation of pharmacotherapy in relation to basic groups of medications (ACEI, statins, P2Y_12_ receptor inhibitors) and non-adherence determinants in one-year follow-up in post-MI patients.

## 2. Tools and Methods

The study was conducted between May 2015 and July 2016 as a single center cohort clinical trial with one-year follow-up, after receiving approval from the local ethics committee of the Collegium Medicum of Nicolas Copernicus University in Toruń (approval number KB312/2015). The study enrolled consecutive patients hospitalized due to myocardial infarction, using the following inclusion criteria: age above 18 years and myocardial infarction treated with percutaneous coronary intervention (PCI). The diagnosis of ST-elevation myocardial infarction (STEMI) and non-ST-elevation myocardial infarction (NSTEMI) was established according to the third universal definition of myocardial infarction [15]. During hospitalization, all the patients enrolled in the study received standard treatment with aspirin, P2Y_12_ receptor inhibitors, statins, ACE inhibitors and beta-blockers. All the participants gave their written informed consent before study enrolment. The following exclusion criteria were applied: contraindications to any of the analyzed medications (ACEI, P2Y_12_ receptor inhibitors or statins); the presence of comorbidities forcing temporary or permanent discontinuation of any of the analyzed medications; a predicted lifespan of less than one year; impaired contact with the patient precluding their active participation in educational interventions. The study was intended to reflect “real world” practice. All the study participants received in-hospital educational interventions on ischemic heart disease, focusing on its symptoms and management. The first educational visit was carried out within the first two days after admission to hospital. The visit included an assessment of patient knowledge of symptoms, knowledge about the disease and knowledge about prevention (20 standardized questions). Then, during the interview, the patient’s doubts were clarified and an educational brochure entitled “Myocardial Infarction” was handed out. Additionally, the patient was motivated to use this source of information during the hospital stay and after discharge. Throughout the entire hospital stay, patients had the opportunity to ask questions and obtain comprehensive answers. The second educational visit, combined with a standardized assessment of readiness for discharge (subjective and objective assessment of the patient’s knowledge and expectations), took place on the last day of hospital stay. Both visits were conducted by trained educational nurses.

Next, after explaining the legitimacy of therapy to the patient, the therapeutic team together with the patient developed the final therapeutic plan, including recommendations for pharmacotherapy and lifestyle modifications.

The analysis of medication discontinuation was performed based on prescription filling data provided by the National Health Fund (NHF) for reimbursed drugs: ACEI (ramipril, perindopril) P2Y_12_ receptor inhibitors (clopidogrel) and statins (atorvastatin, simvastatin, rosuvastatin). The NHF is the only institution in Poland that covers the costs of hospitalization, outpatient treatment and prescribed medications. It is not possible to purchase any reimbursed drug without a prescription registered by the NHF, therefore the data from this source should be considered reliable. Medications non-reimbursed by the NHF (e.g. beta blockers) were not included in the analysis.

The following forms of pharmacotherapy discontinuation were monitored over the period of one year: lack of post-discharge therapy initiation; short-term therapy discontinuation (shorter than 30 days); long-term therapy discontinuation (equal to or longer than 30 days); and chronic cessation of therapy. Lack of post-discharge therapy initiation was defined as failure to fill a prescription received at discharge from hospital, regardless of whether the patient was taking the drug before the index hospitalization. The average duration of discontinuation in individual quarters of follow-up and the day of first discontinuation were assessed. The analysis also included patients who failed to fill prescriptions received at discharge from hospital, which subsequently resulted in a lack of post-discharge therapy initiation with the prescribed medications. Each of the medications was analyzed individually; additionally, a common analysis for all three medications together was performed. In the latter, the day of first discontinuation was defined as the day of the discontinuation of any of the medications. All the enrolled patients received appropriate prescriptions (always including ACEI, P2Y_12_ receptor inhibitors and statins) at discharge from hospital, as patients with contraindications for any of the analyzed medications were excluded from the study. Moreover, follow-up visits were conducted in our hospital in order to ensure appropriate control over timely drug prescription after discharge.

Three hundred seventy-nine patients met the study inclusion criteria, however 127 of them refused to consent to participate in the study, mainly due to the need for follow-up visits in our hospital required by the study protocol. Two hundred fifty-two patients gave their written informed consent and were enrolled in the study. Nevertheless, the analysis was conducted for 225 participants (73.3% men, 26.7% women) aged 30–91 years (mean age 62.9 ± 11.9 years), for whom we obtained data from the National Health Fund. The characteristics of the final study population are presented in Table 1. Due to incomplete data regarding some of the study medications (prescription filling of non-refundable medications was not registered), the final analysis included: 210 patients using ACEI (93.3%), 194 patients on P2Y_12_ receptor inhibitors (86.2%) and 222 patients using statins (98.7%). Taking into consideration the variability of group sizes, the common analysis was conducted for 180 patients (80.0%). In eight cases (3.6%), the duration of follow-up was shortened due to premature death before one year of study enrolment.

Statistical analysis was performed using the Statistica 13.0 package (TIBCO Software Inc, California, USA). Continuous variables were presented as means with standard deviations. The non-normal distribution of data was confirmed by the Shapiro–Wilk test. Therefore, for statistical analysis the nonparametric Kruskal–Wallis one-way analysis of variance was used. Categorical variables were expressed as numbers and percentages. Categorical variables were compared using the χ2 test. To analyze incidences of treatment discontinuation during one-year observation, survival analyses were performed using the Kaplan–Meier method and the log-rank test. To identify predictor variables for failure to start therapy, univariate and multivariate logistic regression models were used. Variables with a *p* value < 0.1 in the univariate analysis were introduced into the multivariate logistic regression model. In order to select the best model, the stepwise backward regression method was applied. The likelihood of failure to start therapy associated with investigated variables was expressed as the odds ratio (OR) and 95% confidence intervals (95%CI). For multivariate analysis, the Cox proportional-hazards model was used. The best multivariate model was obtained using the backward stepwise method. The likelihood of therapy discontinuation associated with investigated variables was expressed as the relative hazards (HR) and 95% CI. Results were considered significant at *p* < 0.05.

## 3. Results

Taking into account periods of therapy discontinuation during the one-year duration of follow-up, the adherence level was 64.1 ± 24.5% for all three groups of medications, 67.2 ± 31.8% for ACEI, 61.6 ± 34.2% for P2Y_12_ receptor inhibitors and 64.4 ± 32.1% for statins.

### 3.1. The Proportion of Patients with Therapy Discontinuation

Table 2 presents the proportions of patients who discontinued therapy with the individual study medications. The highest percentage of patients with any form of therapy discontinuation, long-term discontinuation (≥30 days) or permanent therapy cessation was found in patients taking statins.

### 3.2. Average Duration of Therapy Discontinuation

The average duration of any form of therapy discontinuation over one year of follow-up was 56.1 ± 62.2 days. The analysis of therapy discontinuation according to classes of medications is presented in Figure 1.

### 3.3. The Likelihood of Therapy Discontinuation according to Classes of Medications

For each of the four analyzed forms of therapy discontinuation, the lowest likelihood of discontinuation was found for ACEI and the highest for statins (Figure 2). When analyzing the occurrence of any discontinuation of therapy, significant differences were already seen in the first quarter of follow-up (Figure 2). Starting from the third quarter of follow-up, significant differences were found between individual classes of medications regarding the short- and long-term discontinuation of therapy as well as the permanent cessation of therapy (Figure 2).

### 3.4. Determinants of Lack of Post-Discharge Therapy Initiation

According to the univariate logistic regression analysis, patients with secondary or higher education (OR 3.29, 95% CI 1.23–8.81 *p* = 0.00178) as well as those who are professionally active (OR 2.56, 95% CI 1.04 – 6.34 *p* = 0.0416) have a higher likelihood of a lack of post-discharge therapy initiation with P2Y_12_ receptor inhibitors. The likelihood of the lack of post-discharge therapy initiation with statins was higher in patients remaining professionally active (OR 3.42, 95% CI 1.02–11.45 *p* = 0.0466) and those with a prior MI (OR 3.27, 95% CI 1.05–10.14 *p* = 0.0406). Patients older than 65 years were less likely to remain without therapy with any of the three analyzed medications (OR 0.43, 95% CI 0.199–0.94 *p* = 0.0332) in comparison with their younger counterparts. Univariate logistic regression analysis failed to identify predictors of lack of post-discharge therapy initiation with ACEI or all three medications together.

Multivariate logistic regression analysis identified occupational activity (OR 5.15, 95% CI 1.42–18.65 *p* = 0.013) and a prior MI (OR 5.02, 95% CI 1.49–16.89 *p* = 0.009) as independent predictors of a lack of post-discharge therapy initiation with P2Y_12_ receptor inhibitors. We found no predictors of lack of post-discharge therapy initiation with other medications either when analyzed individually or together.

### 3.5. Determinants of Discontinuation and Cessation of Therapy

Beginning from the first quarter of follow-up, the likelihood of therapy discontinuation was higher in patients older than 65 years (*p* = 0.0003). By contrast, the likelihood of the permanent cessation of therapy was higher in younger patients (*p* = 0.0413) (Figure 3 and Figure 4).

The likelihood of the occurrence of any discontinuation of therapy starting as early as the first quarter of follow-up was higher in unemployed patients (*p* = 0.0003) and those with a history of prior MI (*p* = 0.0042) or cardiac revascularization (*p* = 0.0015) (Figure 3). Patients with a prior MI (*p* = 0.0288) and those with prior cardiac revascularization (*p* = 0.0458) were more likely to permanently quit therapy (Figure 4).

Multivariate analysis indicated age above 65 years (HR −1.59 (95% CI 1.15–2.19) *p* = 0.0049) and prior revascularization (HR −1.44 (95% CI 1.04–2.19) *p* = 0.0273) as independent predictors of therapy discontinuation, but failed to identify independent predictors of the permanent cessation of therapy with any of the medications as well as the temporary discontinuation and permanent cessation of treatment with all three medications together.

## 4. Discussion

The discontinuation and cessation of therapy remains a challenge for therapeutic teams. Its elimination might largely improve the clinical and economical outcomes of treatment [8,16]. Recognizing the importance of this problem, our team have conducted a wide spectrum research in this field [6,17,18,19,20,21].

To the best of our knowledge, this study is the first one to comprehensively analyze the occurrence of nonadherence to post-MI pharmacotherapy, including the incidence of the short- and long-term discontinuation of therapy with different classes of medications and its determinants.

We found that patients are most prone to discontinue therapy between the 2nd and 3rd quarter of follow-up. Previous studies concerning only a single class of medications, usually antiplatelets, have also indicated that therapy discontinuation most commonly occurs at this point in time [4,9,19,22,23,24,25]. In this study, besides the permanent cessation of therapy, we also analyzed the occurrence of the short- and long-term discontinuation of therapy. It is worth noting that, in contrast to the permanent cessation of therapy, a significant increase in the incidence of any form of therapy discontinuation was already seen in the 1st quarter of follow-up.

The observed adherence to medication after discharge was consistent with previously reported data. In a study published by Latry et al. [26], the discontinuation rate of dual antiplatelet treatment for at least 1 month was 18.6% during the first 3 months and 49.1% at 12 months in patients treated with PCI for acute myocardial infarction. As reported by Choudhry et al. [27], during one-year follow-up the mean adherence to therapy with statins and ACEI/ARB was 68.6% and 66.4%, respectively. In a study by Kirchmayer et al. [28], in one-year follow-up an adherence rate of ≥80% was reported for 76.1% of patients on statin therapy and 64.4% of patients receiving ACEI/ARB. In their study evaluating adherence to treatment during the first year after MI, Jánosi et al. [29] reported an adherence rate of ≥80% in 54.4% of patients receiving statins and 64.0% of patients treated with ACEI/ARB. Our study also showed that adherence to treatment expressed as the average duration of therapy discontinuation in one year follow-up was lowest for statins (75 ± 72.2 days without medication), as compared with ACEI (47.6 ± 52.0 days without medication) and P2Y_12_ receptor inhibitors (45.6 ± 49.1 days without medication). On the other hand, the highest prevalence of a lack of post-discharge therapy initiation was found for P2Y_12_ inhibitors (11.3%). Dayoub et al. [30], reporting data from a large US national insurer, showed an even higher proportion of patients (19.3% in 2016) who did not fill any P2Y_12_ inhibitor prescription within 30 days of discharge.

According to our results, among the analyzed sociodemographic factors, age, occupational status, and the level of education are important determinants of lack of post-discharge initiation or discontinuation of therapy. We also observed some interesting relations between adherence and patient age. Elderly patients had a high likelihood of lack of post-discharge therapy initiation or discontinuation of therapy with any of the medications; however, at the same time there was a lower likelihood of permanent cessation of therapy in comparison with younger patients. The relevant literature shows inconsistent data. Degli Esposti et al. found a lower adherence to statin therapy in younger patients [31], while Zhu et al. reported the same with respect to P2Y_12_ receptor inhibitors [32]. On the contrary, Kang et al. found a higher adherence to antihypertensive treatment in younger patients [33]. D’Ascenzo et al. [34] indicate that older age is a predisposing factor for bleeding during antiplatelet therapy, which may translate into more frequent discontinuation of treatment in this group of patients. By contrast, Naderi et al. [4] as well as Wong et al. [35] demonstrated no relationship between age and adherence.

In our study, we also noticed differences regarding a lack of post-discharge therapy initiation, any discontinuation of therapy and permanent cessation of therapy in relation to patient employment status. According to previous studies, unemployed persons usually have a lower adherence to pharmacotherapy after MI [17,22,32,33,35,36,37]. Our study also showed that the likelihood of therapy discontinuation was higher in unemployed patients. However, against our expectations, we found that employed patients had a higher likelihood of lack of post-discharge therapy initiation with P2Y_12_ receptor inhibitors and statins. In contrast to previous studies reporting higher adherence levels in patients with secondary and higher education [9,36], we found that those patients have a higher likelihood of lack of post-discharge therapy initiation with P2Y_12_ receptor inhibitors.

Unexpectedly, we also found a higher likelihood of lack of post-discharge therapy initiation with statins and the temporary discontinuation of therapy and permanent cessation of therapy in patients with a prior MI and cardiac revascularization. However, we consider pointing to prior MI, PCI or coronary artery bypass grafting (CABG) in this context as predisposing factors for therapy discontinuation a misinterpretation. On the contrary, therapy discontinuation and low adherence following previous interventions might consequently lead to MI, the latter being the reason for inclusion in our study [17]. This would not be a solitary case. Suzuki et al. [22] identified previous revascularization as a predisposing factor for lower adherence and more frequent therapy discontinuation. There are also reports indicating that patients with no history of cardiac disease have higher adherence levels [17,32,37,38].

A limitation of this study that needs to be mentioned is the fact that patients who received medications non-reimbursed by the National Health Fund were excluded from the analysis. We also only analyzed a limited number of factors as potential determinants of therapy discontinuation; we did not include factors such as patient socio-economic status and knowledge or presence of side effects. Moreover, we do not have patients’ reports concerning the reasons for therapy discontinuation. On the other hand, the strengths of this study are its comprehensiveness and the homogeneity of the study population.

## 5. Conclusions

The majority of post-MI patients discontinue, either temporarily or permanently, therapy with one of the essential classes of medications during the first year after myocardial infarction. The most common medication class to be discontinued are statins. Older age and prior cardiac revascularization are independent determinants of therapy discontinuation.

## Figures and Tables

**Figure 1 jcm-09-04109-f001:**
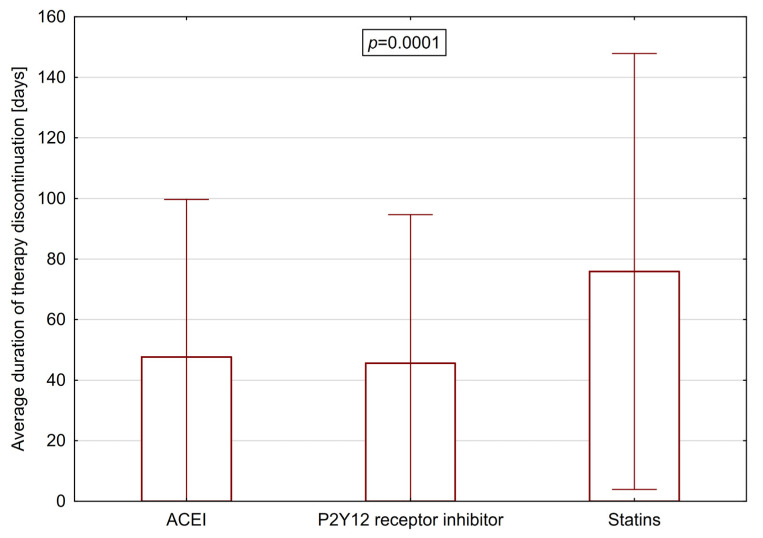
Average duration of therapy discontinuation according to classes of medications in one-year follow-up.

**Figure 2 jcm-09-04109-f002:**
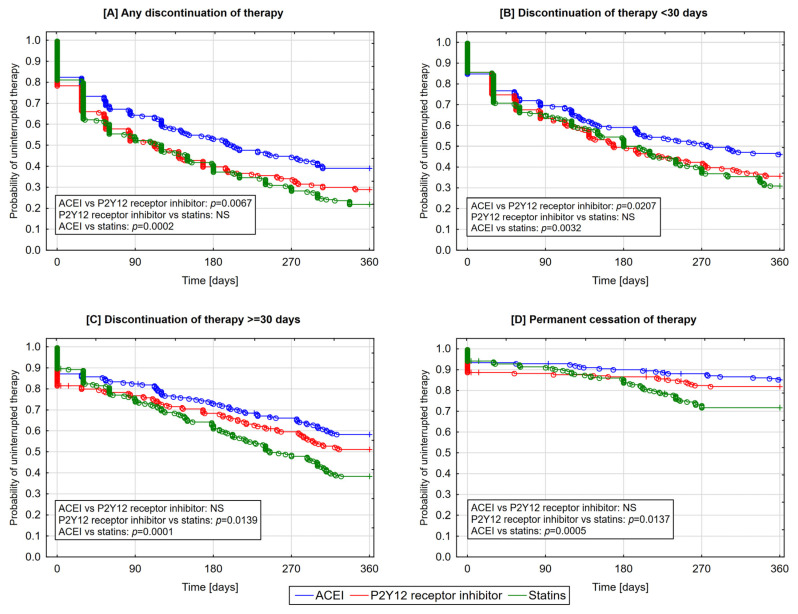
Cumulative incidence of uninterrupted therapy with individual classes of medications—Kaplan–Meier curves (**A**—any discontinuation of therapy; **B**—discontinuation of therapy <30 days; **C**—discontinuation of therapy ≥30 days; **D**—permanent cessation of therapy). ○—complete follow-up; +—truncated follow-up.

**Figure 3 jcm-09-04109-f003:**
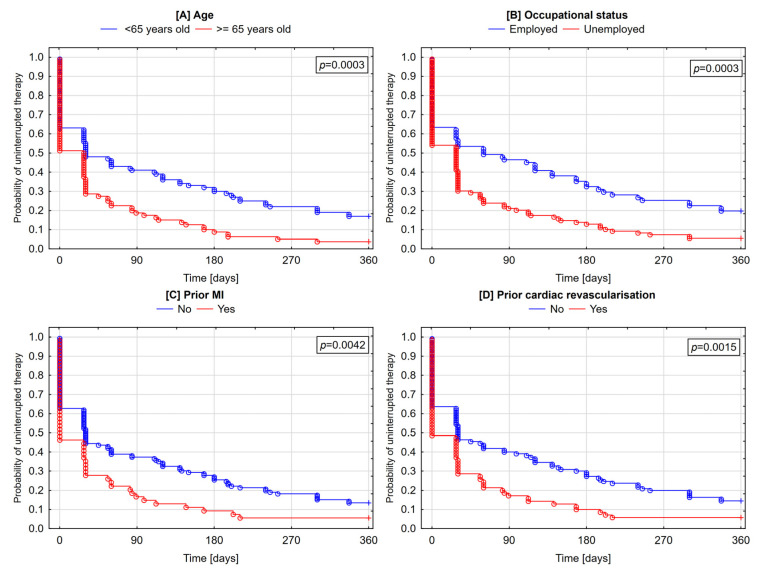
Cumulative incidence of uninterrupted therapy with all three medications—Kaplan–Meier curves (**A**—age; **B**—occupational status; **C**—prior MI; **D**—prior cardiac revascularization). ○—complete follow-up; +—truncated follow-up.

**Figure 4 jcm-09-04109-f004:**
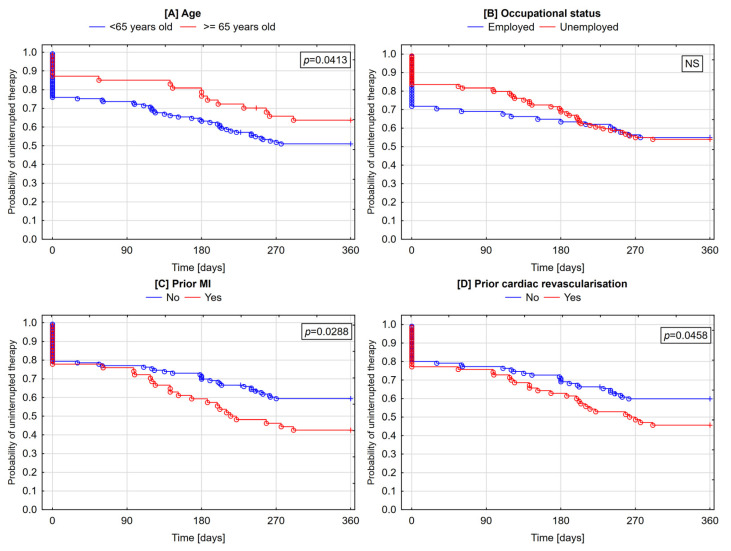
Cumulative incidence of lack of therapy cessation with all three study medications—Kaplan–Meier curves (**A**—age; **B**—occupational status; **C**—prior MI; **D**—prior cardiac revascularization). ○—complete follow-up; +—truncated follow-up.

**Table 1 jcm-09-04109-t001:** Characteristics of the study population.

Parameter	Variable	Total Sample	
		*n*	%
Gender	Female	60	26.7
	Male	165	73.3
Age	<65	129	57.3
	≥65	96	42.7
Employment status	Employed	93	41.3
	Unemployed	132	58.7
Education	Primary	30	13.3
	Vocational	83	36.9
	Secondary	82	36.4
	Higher	30	13.3
Self-reported economic status	Very good	14	6.2
	Satisfactory	199	88.4
	Bad	12	5.3
	Very bad	0	0
Marital status	Unmarried	25	11.1
	Widowed	33	14.7
	Married	167	74.2
Place of residence *	City	117	52.0
	Town	44	19.6
	The country	64	28.4
History of CAD	Yes	102	45.3
	No	123	54.7
Prior MI	Yes	64	28.4
	No	161	71.6
Prior cardiac revascularization	Yes	85	37.8
	No	140	62.2
Hyperlipidemia	Yes	151	67.6
	No	73	32.4
Diabetes	Yes	63	28.0
	No	157	71.0
Hypertension	Yes	165	73.3
	No	60	26.7
Smoking status (current)	Yes	85	37.8
	No	140	62.2

* City > 100,000 inhabitants; town ≤ 100,000 inhabitants. CAD—coronary artery disease. MI—myocardial infarction.

**Table 2 jcm-09-04109-t002:** Proportion of patients with therapy discontinuation according to classes of medications.

Form of Therapy Discontinuation	ACEI *N* = 210 [%]		P2Y_12_ Receptor Inhibitors *N *= 194 [%]		Statins *N* = 222 [%]		*p* *	ACEI + P2Y_12_ Receptor Inhibitors + Statins (Discontinuation of Any Medication) *N* = 180 [%]		ACEI + P2Y_12_ Receptor Inhibitors + Statins (Simultaneous Discontinuation of All Medications) *N* = 180 [%]	
	No	Yes	No	Yes	No	Yes		No	Yes	No	Yes
Any discontinuation of therapy	39.1	60.9	28.9	71.1	22.1	77.9	**0.0006**	12.2	87.8	92.8	7.2
Short-term discontinuation of therapy (<30 days)	56.7	43.3	53.1	46.9	50.5	49.5	0.4310	15.6	84.4	96.7	3.3
Long-term discontinuation of therapy (≥30 days)	56.6	41.4	51.5	48.5	37.7	61.3	**0.0002**	16.6	83.9	96.2	3.8
Permanent cessation of therapy	85.2	14.8	82.0	12.8	72.1	27.9	**0.0001**	55.0	45.0	97.8	2.2
Lack of post-discharge therapy initiation	93.3	6.7	88.7	11.3	94.1	5.9	0.0860	75.1	24.1	98.3	1.7

* angiotensin converting enzyme inhibitors ACEI vs. P2Y_12_ receptor inhibitors vs. statins.

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
