# Peer review of "Therapy Discontinuation after Myocardial Infarction"

_jcm, 2020, doi:10.3390/jcm9124109_

Round 1
Reviewer 1 Report
This is a study about therapy discontinuation after myocardial infarction. Overall the study seems to be really confused, starting from the inclusion criteria. Moreover it seems to be really difficult to have so high percentage of therapy discontinuation in a population of young patients. Are results related to a difficult management of patients discharged with a diagnosis of MI? or are results related to huge methodologic gap?
Which is the definition of MI used?
Why it was not assessed the use of beta-blockers?
It is not clear if patients need to receive therapy with ACE inhibitors before the index hospitalization for MI.
What does it mean that authors valued “lack of therapy initiation?”. If patients enrolled were myocardial infarction treated with PCI it is really hard to understand how they cannot initiate therapy with P2Y12 inhibitors.
How is it possible to reach a 50% of discontinuation of P2Y12 inhibitor at 30 days in a population of patients with a mean age 62 years old.
Reviewer 2 Report
The presented manuscript reports an analysis of incidence and prevalence. This is of course a worthy and often overlooked topic. It’s certainly no use coming up with drug regimens that are theoretically efficacious if patients are not going to take them. New data in this area are welcome and the manuscript is well written.
However, I have the following points which I feel should be clarified in any revision of the manuscript.
- Was the in-hospital educational intervention part of standard care or specific to this study? Is this study intended as one of ‘real world’ practice or an evaluation of the educational intervention?
- How many patients were approached about the study but declined to consent? Whilst taking a prospective approach is typically regarded as more robust than a retrospective approach in these kinds of studies, this factor might significantly skew the representation of certain patient groups or attitudes within the study sample.
- I am slightly surprised too by the finding that therapy was less likely to be initiated in the more educated, employed and those with prior MI. The authors need to provide more data on the numbers of patients in each group that did not initiate therapy and reasons for this – how many were prescribed appropriate therapy but did not fill the prescription, for example? Was the decision not to initiate appropriate therapy on the part of the Doctor, the patient or both?
- When the investigators say ‘initiated’ does this include or not include those who were already receiving a relevant medication before the index hospital episode and who continued to do so?
- Could the authors either provide a breakdown of discontinuation by specific agent (in particular by specific P2Y12 inhibitor) or failing this outline the proportion that received each specific agent, or in turn failing this provide a summary of the standard agents used in their centre.
- How was Economic status collected, was this self reported by the participants or obtained by some other method?
- I presume the authors do not have data concerning the reason for discontinuation, but if they do this should be included too.
- The ’12’ in ‘P2Y12’ should be in subscript throughout.
I think these points need to be addressed before the manuscript can be considered further.
Round 2
Reviewer 1 Report
Authors replied to the majority of questions raised. I still have concerns about the high degree of therapy discontinuation above all for P2Y12 inhibitors. Thus a more specific description on education of patients received during hospital stay is needed. Next is it not possible that patients continued P2Y12 inhibitors without using National Health Found? Is national health found reliable enough for drug there prescription? Are there any ways patients have to obtain drugs as P2Y12 inhibitors?
